# Effects of scapular-focused movement-based exercises on sports performance of athletes with scapular dyskinesis: A systematic review

Mònica Solana–Tramunt[1], Hossein Fakoor Rashid[2], Narges Norouzi[3],
Yaser Dehghan[4], Hossein Khazanin[2], Bahareh Sadegh[5], Mohammad Alimoradi [6,7],
Hassan Daneshmandi[2], Mohammad Alghosi [8]*

1 Department of Sports Sciences and Physical Activity, Faculty of Psychology, Education and Sport Sciences Blanquerna (FPCEE Blanquerna), Ramon Llull University, Research Group in Health, Physical Activity and Sport (SAFE), Barcelona, Spain, 2 Department of Sports Injury and Corrective Exercise, Faculty of Physical Education and Sport Sciences, University of Guilan, Rasht, Iran, 3 Faculty of Sport Sciences, Alzahra University, Tehran, Iran, 4 Department of Physical Education, PayameNoor University, Tehran, Iran, 5 Department of Sport Injury and Corrective Exercises, Faculty of Sports Science, University of Isfahan, Isfahan, Iran, 6 Department of Sports Injuries and Corrective Exercises, Faculty of Sports Science, Shahid Bahonar University of Kerman, Kerman, Iran, 7 HERC- Health, Exercise & Research Center, Mina Rashid, Dubai Maritime City, Dubai, United Arab Emirates, 8 Department of Physical Education, Technical and Vocational University (TVU), Tehran, Iran

* mohammadalghosi9@gmail.com

## Abstract

### Background

Scapular dyskinesis is a common dysfunction among athletes, particularly in overhead sports, leading to pain, reduced range of motion (ROM), and impaired performance. Movement-based exercises are increasingly used to address these issues, but their overall impact on sports performance remains unclear.

### Objective

This systematic review aims to evaluate the effects of movement-based exercises on sports performance in athletes with scapular dyskinesis.

### Method

A comprehensive search was conducted in Web of Science, Scopus, and PubMed up to July 30, 2025, following PRISMA guidelines. Data were extracted and assessed for risk of bias using RoB-2 and ROBINS-I tools. A narrative synthesis was performed due to study heterogeneity.

### Results

Fourteen studies (8 RCTs and 6 non-RCTs) involving 412 participants with a mean age of 23.8 years assessed movement-based interventions lasting from a single

**Data availability statement:** All relevant data are within the paper and its Supporting information files.

**Funding:** The author(s) received no specific funding for this work.

**Competing interests:** The authors have declared that no competing interests exist.

session to 24 weeks, primarily focusing on scapular stabilization, kinetic chain control, and proprioception over 6–8 weeks with around three sessions per week. Moderate-certainty evidence suggests that exercise likely improves shoulder function, disability, and glenohumeral range of motion over 6–12 weeks. However, the evidence for pain reduction and improvement in rotator cuff/scapular strength is of low certainty, showing mixed effects depending on the specific program. Evidence for improvement in scapular kinematics is also of low certainty. Sport-specific performance outcomes, such as throwing velocity, remain highly uncertain due to small sample sizes and conflicting results from RCTs.

## Conclusion

Movement-based exercises may be considered by athletes with scapular dyskinesis to potentially improve shoulder function and glenohumeral range of motion; however, the certainty of evidence for effects on pain relief, strength, and sports performance is very low. Therefore, strong recommendations cannot be made at this stage. More tailored programs and well-structured RCTs are needed to clarify these effects.

## 1. Introduction

Scapular dyskinesis, characterized by abnormal scapular motion during shoulder movements, is a prevalent condition among athletes, particularly those engaged in overhead sports such as baseball, volleyball, and tennis [1–3]. Epidemiological studies indicate that the prevalence of scapular dyskinesis varies considerably by population, ranging from 42% in asymptomatic athletes to 81% in symptomatic athletes, underscoring its common presentation and clinical relevance in sports medicine [4,5]. This dysfunction is associated with compromised shoulder mechanics, which may result in pain, decreased range of motion (ROM), and potentially impaired athletic performance [6,7]. The scapula is a critical link in the upper extremity kinetic chain, providing a stable base for glenohumeral rotation and facilitating optimal force transfer from the core and lower body to the throwing or striking arm [8–10]. Classic biomechanical models posit that scapular dyskinesis impairs this role, leading to a disruption in the kinetic chain [2,9,10]. Specifically, altered scapular position and motion, such as excessive anterior tilt or internal rotation, compromise glenohumeral stability and subacromial space, which is frequently associated with shoulder pain and pathology like impingement and rotator cuff tendinopathy [6,7,11–13]. Furthermore, this dysfunction is theorized to directly hinder sports performance by impairing the efficient generation and transfer of forces, potentially resulting in decreased throwing velocity, reduced shot power, and diminished accuracy [9,10]. However, it is important to acknowledge that direct, high-level evidence linking the specific correction of scapular dyskinesis to measurable gains in these sports performance outcomes remains scarce, representing a significant gap in the current literature. Athletes with scapular dyskinesis often report symptoms such as shoulder pain, weakness, and functional limitations, which may be associated with reduced training

and competitive performance [3,14]. Recent studies have highlighted the importance of addressing scapular dyskinesis through targeted rehabilitation strategies [15,16]. Given the increasing physical demands in competitive sports, early recognition and effective management of scapular dyskinesis are critical for injury prevention and performance optimization [17]. Movement-based exercises, which focus on enhancing scapular stability, strength, and coordination, have emerged as a promising intervention for athletes suffering from this condition [18–20]. Unlike passive modalities or surgical interventions, movement-based approaches directly target neuromuscular control and kinetic chain deficits, making them particularly relevant for athletes aiming to return to sport [21,22]. These exercises aim to restore normal scapular mechanics, which may help improve shoulder function and overall sports performance [15,23–25]. Various modalities, including resistance training, proprioceptive exercises, and neuromuscular re-education, have been employed to target the underlying deficits associated with scapular dyskinesis [26–28]. Despite the growing body of literature, the existing syntheses addressing the impact of movement-based exercises on sports performance in athletes with scapular dyskinesis remain limited in scope and consistency. Although previous reviews have partially explored performance-related or clinical aspects [14,29], their primary emphasis has largely been on clinical and biomechanical outcomes rather than on systematic evaluation of sport-specific performance measures.

Most prior reviews have emphasized clinical outcomes such as pain and disability, but few have systematically examined whether these improvements translate into measurable changes in sports performance parameters, such as throwing velocity, agility, and upper-limb stability [22,30]. Previous reviews have often focused on specific exercise protocols or individual outcomes, leaving uncertainty regarding the broader and comparative implications of different movement-based interventions on performance-related variables [15,26]. This systematic review aims to fill this gap by critically evaluating and summarizing the findings of relevant studies. By examining the effects of various movement-based interventions on key performance metrics, including pain reduction, ROM, muscle strength, and functional outcomes, this review seeks to provide a clearer understanding of the potential associations between these exercises and improvements in key performance metrics in athletes with scapular dyskinesis.

## 2. Methods

### 2.1 Protocol registration

This systematic review was conducted in accordance with the Preferred Reporting Items for Systematic Reviews and Meta-Analyses (PRISMA) [31] (S1 File) and is registered with the International Prospective Register of Systematic Reviews (PROSPERO) under the registration number CRD420251135347.

### 2.2 Search strategy

A thorough, systematic search was conducted across the Web of Science, Scopus, and PubMed databases from their inception to July 30, 2025. Two independent reviewers, MALG and MALI, carried out the searches, with any disagreements resolved through discussion or, when necessary, consultation with a third reviewer (HD). The search strategy combined MeSH terms and free-text keywords using both AND and OR operators to ensure comprehensive coverage. The specific search strategies are detailed in Table 1. No restrictions were applied regarding study design or publication status. To supplement the database search, additional relevant studies were identified through manual screening of reference lists from included articles and eligible reviews. Broader literature discovery was supported by searches in Google Scholar and the Connected Papers platform (https://www.connectedpapers.com). These supplementary methods were implemented to enhance the retrieval of pertinent publications, including grey literature.

### 2.3 Eligibility criteria and study selection

After completing the search, all identified studies were imported into EndNote Reference Library (Version 20; Clarivate Analytics, Thomson Reuters Corporation, Philadelphia, Pennsylvania) where duplicates were meticulously identified

**Table 1. Search strategy for each database.**

| Database | Complete search strategy |
|---|---|
| Web of Science | athlet* OR sport* OR sportsman* OR sportswoman* OR sportsperson* (Topic) AND exercis* OR train* OR rehabilitat* OR physiotherap* OR "therapeutic exercise" (Topic) AND perform* OR strength* OR mobil* OR "range of motion" OR endurance OR function* OR skill* OR agility OR speed OR power OR injur* OR fatigue OR strength (Topic) AND "scapular dyskinesis" OR OR scapul* (Topic) |
| Scopus | (TITLE-ABS-KEY (athlet* OR sport* OR sportsman* OR sportswoman* OR sportsperson*) AND TITLE-ABS-KEY (exercis* OR train* OR rehabilitat* OR physiotherap* OR "therapeutic exercise") AND TITLE-ABS-KEY (perform* OR strength* OR mobil* OR "range of motion" OR endurance OR function* OR skill* OR agility OR speed OR power OR injur* OR fatigue OR strength) AND TITLE-ABS-KEY ("scapular dyskinesis" OR scapul*)) |
| PubMed | (((athlet*[Title/Abstract] OR sport*[Title/Abstract] OR sportsman*[Title/Abstract] OR sportswoman*[Title/Abstract] OR sportsperson*[Title/Abstract]) AND (exercis*[Title/Abstract] OR train*[Title/Abstract] OR rehabilitat*[Title/Abstract] OR physiotherap*[Title/Abstract] OR "therapeutic exercise"[Title/Abstract])) AND (perform*[Title/Abstract] OR strength*[Title/Abstract] OR mobil*[Title/Abstract] OR "range of motion"[Title/Abstract] OR endurance[Title/Abstract] OR function*[Title/Abstract] OR skill*[Title/Abstract] OR agility[Title/Abstract] OR speed[Title/Abstract] OR power[Title/Abstract] OR injur*[Title/Abstract] OR fatigue[Title/Abstract] OR strength[Title/Abstract])) AND ("scapular dyskinesis"[Title/Abstract] OR scapul*[Title/Abstract]) |

and removed. The refined list of studies was then transferred to the Rayyan web application (Rayyan Systems, Inc., Cambridge, MA, USA) [32] for screening. Within Rayyan, each study underwent a detailed evaluation based on its title, abstract, and, when necessary, its full text. Any studies deemed irrelevant during this preliminary screening had their exclusion reasons carefully recorded. Two independent reviewers, YD and BS, subsequently conducted a thorough full-text review of studies that met initial criteria, applying the PICOS framework [33] (Population, Intervention, Comparison, Outcome, and Study design) to guide inclusion and exclusion decisions, as outlined in Table 2. For clarification, studies reporting either pain, performance, or both outcomes were considered eligible. Single-session (acute) interventions were also included to provide insight into immediate effects. These clarifications do not alter the studies included in this review. When disagreements occurred during this process, a third reviewer, MST, was consulted to reach a consensus on whether to include or exclude the contested studies.

## 2.4 Data extraction

From the included papers, study details (author, year of publication, location), study design, sample description (sample size, sex, age, and scapula dyskinesis type), exercise characteristics of experiment and control group, sports performance measures, and main outcomes position were extracted by two authors (HK and MALI). If any important information was missing, the corresponding authors were contacted via email, with a maximum of three attempts made to obtain the necessary details (n = 0).

## 2.5 Quality assessment

The assessment of risk of bias in the included studies was conducted in accordance with Cochrane guidelines. Specifically, the ROBINS-I tool was employed to evaluate non-randomized controlled trials (non-RCTs), while the RoB-2 tool was used for randomized controlled trials (RCTs). Two reviewers, HK and NN, independently appraised the studies by examining all relevant domains: the seven domains covered by ROBINS-I, including confounding, selection of participants, classification of interventions, deviations from intended interventions, missing outcome data, measurement of outcomes, and selection of reported results; and the five domains of RoB-2, with an additional focus on bias related to the randomization process for RCTs. Both tools provided structured guidance and signalling questions to support transparent and consistent judgments, with categories such as "Low risk of bias," "High risk of bias," "Some concerns", "Moderate," and "Unclear risk of bias" or "No information," alongside detailed explanatory notes. To visually summarize the findings of the

**Table 2. Eligibility criteria based on PICOS strategy.**

|  | Inclusion criteria | Exclusion criteria |
|---|---|---|
| Population | Athletes with a diagnosis of scapular dyskinesis. Studies including participants with concurrent shoulder conditions were included only if scapular dyskinesis was explicitly diagnosed and targeted by the intervention | Athletes with scapular dyskinesis and concurrent injuries (e.g., rotator cuff tendinopathy, superior labrum anterior to posterior lesions, labral tears). Scapular dyskinesis in non-athletic populations. |
| Intervention | Movement-based Interventions: in which the therapeutic effect is primarily achieved through active, voluntary movement performed by the participant, involving muscle activation, postural control, or neuromuscular retraining, rather than passive techniques | Other interventions are occurring simultaneously. |
| Comparison | A control condition (e.g., no intervention, placebo, usual care, or an alternative non-scapular-focused rehabilitation protocol). | Studies without a control condition. |
| Outcome | Studies were required to report on at least one primary outcome: (1) pain intensity assessed specifically via the Visual Analog Scale or Numeric Rating Scale, or (2) objective sport-performance metrics (e.g., throwing velocity, jump height, electromyographic measures). | Absence of measurements for pain intensity or performance-related outcomes. |
| Study design | RCTs and non-RCTs. | Single-group intervention; Case studies; Reviews. |

**Abbreviations:** Non-RCT: Non-Randomized Controlled Trial, RCT: Randomized Controlled Trial.

bias assessment, a traffic light plot and summary plot were generated using the Robvis visualization tool (www.riskofbias.info) [34], in line with Cochrane's recommendations for clear presentation of risk of bias results.

## 2.6 Data synthesis

This study employed a narrative approach to data synthesis to provide a thorough and transparent account of the findings. To uphold methodological rigour and improve the credibility and trustworthiness of the results, the review strictly adhered to the PRISMA statement guideline [31]. Given the heterogeneity in outcome measures among the included studies, performing a meta-analysis was deemed inappropriate and not possible. Although some outcomes (e.g., IR ROM, ER strength, VAS) may appear conceptually similar, substantial differences in measurement protocols, timing, and participant characteristics made pooling the data inappropriate. Therefore, a narrative synthesis was applied to accurately reflect study-specific results and heterogeneity. Consequently, the review utilized a Synthesis Without Meta-Analysis (SWiM) guideline [35], Exercise intensity was not reported in most studies, so SWiM was applied to qualitatively summarize intervention effects rather than conducting a dose–response synthesis. a method previously employed in systematic reviews [36,37], to ensure clarity and consistency in the presentation of synthesized evidence. Additionally, prior to the data merging, the level of agreement between reviewers at each stage of the evaluation process was systematically assessed using Kappa ($\kappa$) statistics. The strength of agreement was categorized into distinct levels: poor ($\kappa \leq 0.20$), fair ($\kappa = 0.21–0.40$), moderate ($\kappa = 0.41–0.60$), substantial ($\kappa = 0.61–0.80$), or near-perfect ($\kappa = 0.81–0.99$) [38].

## 2.7 Certainty of evidence assessment

The certainty of the evidence for each primary outcome was assessed using the Grading of Recommendations, Assessment, Development, and Evaluation (GRADE) approach [39], as detailed in the Cochrane Handbook for Systematic Reviews of Interventions [40]. The GRADE framework evaluates evidence across five domains: risk of bias, inconsistency, indirectness, imprecision, and publication bias. Outcomes are categorized as having high, moderate, low, or very low

certainty. [39]. Since most of the studies included were RCTs, the initial level of certainty was considered high and down-graded as necessary based on the identified limitations.

## 3. Results

### 3.1 Study identification

Based on the PRISMA guideline [31], an initial search across electronic databases identified 1716 records. After removing 685 duplicates (40.0%), 1031 unique studies remained for screening against the inclusion and exclusion criteria. During the title and abstract review, 1008 studies (97.8%) were excluded as irrelevant, leaving 23 articles (1.3%) for full-text assessment. Of these, 9 studies were excluded for specific reasons outlined in S2 File. Ultimately, 14 studies were included in the systematic review. Inter-rater agreement was perfect ($\kappa = 0.91$), confirming consistent bias assessments across reviewers. These studies specifically examined the impact of movement-based exercise interventions on the sports performance of athletes with scapular dyskinesis. The flow of studies through each screening stage is illustrated in the PRISMA diagram (Fig 1).

### 3.2 Descriptive characteristics of the included studies

The fourteen studies included in this systematic review, published between 2010 and 2025, encompassed 412 participants. Among the 373 participants with reported sex, 242 (64.9%) were male, and 131 (35.1%) were female. Among these studies, eight were randomized controlled trials (RCTs) conducted in Iran (n = 3), Germany (n = 1), Greece (n = 1), the United Kingdom (n = 1), China (n = 1), and South Korea (n = 1). The other six studies were non-randomized controlled

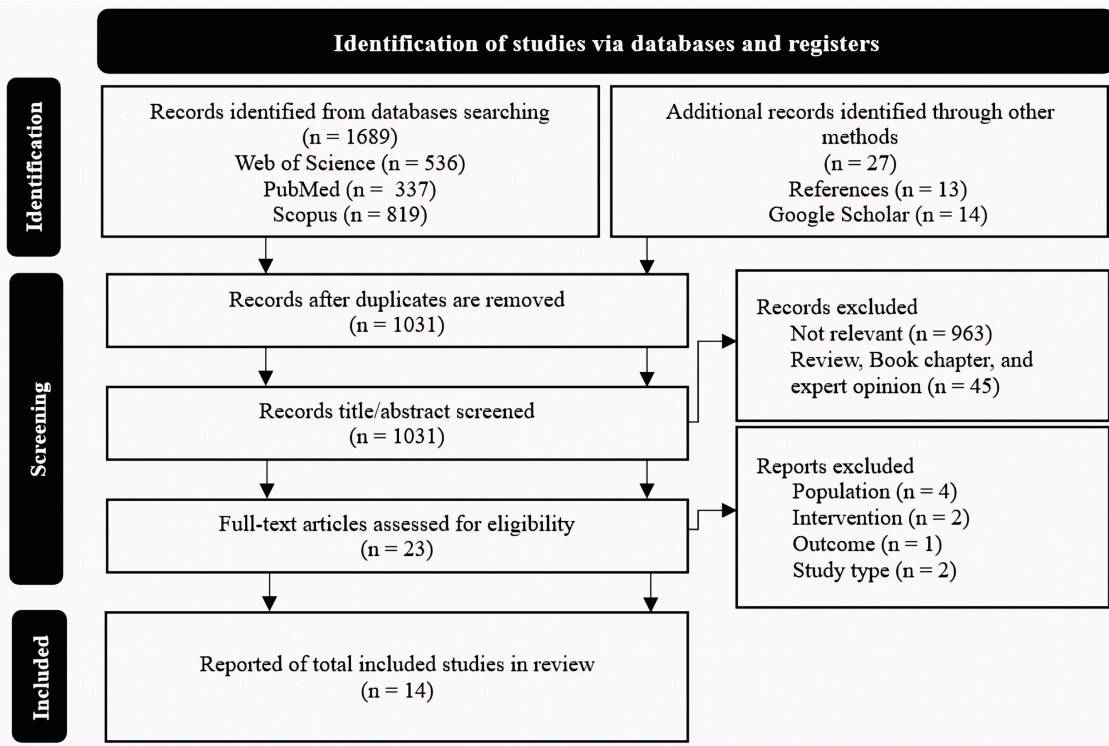

**Fig 1. Preferred Reporting Items for Systematic Reviews and Meta-Analyses (PRISMA) 2020 flow diagram for new systematic reviews, including searches of databases and registers.**

trials conducted in Italy (n = 3), Brazil (n = 1), South Korea (n = 1), and Iran (n = 1). Sample sizes across the included studies ranged from 4 to 54 participants. The pooled mean age of participants was 23.8 ± 3.1 years. Table 3 provides a detailed overview of the characteristics of the included studies.

### 3.3 Effects of exercises on sports performance

This systematic review analyzed studies investigating a wide range of variables related to sports performance, pain, ROM, muscle strength, scapular kinematics, and upper limb function. The overall findings for each category are summarized below:

**3.3.1 Upper limb function and performance.** Multiple studies have reported improvements in upper limb function and stability following exercise interventions. Sant et al. (2018) observed increases in functional stability and upper limb strength after pre-treatment interventions, although no statistically significant differences were found between groups [41]. Meanwhile, in the study by Nowotny et al. (2018), scapula-focused exercises specifically led to greater improvements in shoulder function [42]. Regarding shoulder positioning and specialized functional performance, Karimi and Firouzjah (2024) reported significant improvements in shoulder posture and related functional outcomes following an exercise program, which were statistically superior to the control group [20]. Additionally, Khakpourfard et al. (2023) documented significant enhancements in functional stability and shoulder proprioceptive accuracy post-intervention, indicating improved neuromuscular control [18]. In the domain of throwing performance, Paraskevopoulos et al. (2022) reported significant increases in throwing performance index and velocity in both intervention groups; however, increased throwing strength was observed only in the Motor Control Exercise group [43].

**3.3.2 Range of motion.** Improvement in ROM, particularly for internal rotation, was a common finding across multiple studies, though the effects on external rotation and the overall pattern of adaptation varied. Wen et al. (2025) reported that both intervention groups showed improved active ROM by week 8, but only the Scapular Dyskinesis–Based Exercise Therapy group maintained gains post-intervention [7]. Several studies reported significant gains in shoulder internal rotation ROM [20,23,24,41,44]. The findings of Ilyoung et al. (2018) illustrate a specific adaptive pattern, with a 15° increase in internal rotation ROM concurrently with a reduction in external rotation ROM, resulting in a net decrease in glenohumeral internal rotation deficit [45]. This suggests that some protocols may elicit a targeted shift in rotational balance rather than a uniform increase in all directions.

**3.3.3 Muscle activity and strength.** Several studies have reported significant changes in muscle activity and strength following exercise interventions targeting the shoulder complex. Song et al. (2020) documented a significant increase in activity of the upper and lower trapezius muscles post-intervention. In the SICK-Dominant group, changes in serratus anterior muscle activity were observed before and after training, with these differences persisting after the intervention [46]. Moura et al. (2016) also reported improvements in serratus anterior muscle activation [44].

Regarding muscle strength, Wen et al. (2025) observed increases in isometric strength exclusively in the Scapular Dyskinesis–Based Exercise Therapy group [7]. Similarly, Khakpourfard et al. (2023) reported significant gains in internal and external rotator muscle strength [18]. Moura et al. (2016) documented enhanced strength of shoulder extensors and external rotators [44]. Ilyoung et al. (2018) found a significant increase in peak isokinetic torque of external rotation in the Progressive Scapular Stabilization Exercise group, while the Scapular Stabilization Exercise group showed no significant changes [45]. Furthermore, Merolla et al. (2010), in three independent studies, demonstrated sustained increases in supraspinatus and infraspinatus muscle strength at 3- and 6-month follow-ups post-rehabilitation [23–25].

**3.3.4 Scapular kinematics.** Numerous studies have demonstrated significant improvements in scapular kinematics and scapulohumeral rhythm following exercise interventions. Wen et al. (2025) reported that 43.8% of participants in the Scapular Dyskinesis–Based Exercise Therapy group showed significant improvements in scapular kinematics [7]. Gholamian et al. (2024) noted enhancements in scapulohumeral rhythm and upper limb function after their exercise program [19]. Likewise, Naderifar and Ghanbari (2022) observed a significant reduction in scapular dyskinesis following

**Table 3. Characteristics of the included studies.**

| Study details | Study design | Sample description | Exercise characteristics of EG | CG intervention | Sports performance measures | Main outcomes |
|---|---|---|---|---|---|---|
| Wen et al., (2025) China | RCT | N = 32 Sex = 32 males Age = 20.8 ± 2.4 years SD type = NR | D = 8 weeks F = 3 per week I = NR T = NR T = EG1: Scapular dys-kinesisbased exercise therapy; EG2: Multimodal physical therapy | NR | Pain, ROM, strength, disability index, scapular kinematics | Disability improved in both groups by week 8 (p < 0.001) and remained only in SDBET at week 12 (p < 0.001). Pain was reduced more in MPT at week 8 (p = 0.018) but not at week 12 (p = 0.268). Active ROM improved in both groups by week 8 and remained only in SDBET at week 12 (p < 0.001). Strength improved only in SDBET at weeks 8 and 12 (p < 0.001). Scapular kinematics improved in 43.8% of SDBET participants, with no change in MPT (p = 0.001–0.004). |
| Gholamian et al., (2024) Iran | RCT | N = 30 Sex = 30 males Age = 26.3 ± 1.4 years SD type = NR | D = 8 weeks F = 3 per week I = NR T = NR T = Functional exercises | Regular tennis training and daily activities | Scapular brachial rhythm, upper limb function | The results indicated that functional exercises significantly improved scapulohumeral rhythm at 0° (p = 0.004), 45° (p < 0.001), 90° (p < 0.001), and 135° (p < 0.001), as well as upper limb function (p = 0.002) in the experimental group. |
| Karimi and Firouzjah, (2024) Iran | RCT | N = 30 Sex = 30 females Age = 22.7 ± 2.6 years SD type = NR | D = 8 weeks F = 3 per week I = NR T = 40 minutes T = Scapular stabilization exercises | Usual daily activities | Specific performance, Shoulder position, Pain | Training led to significant improvements in shoulder position and performance (p = 0.001 for both). The control group also showed performance gains at eight weeks. After adjusting for pre-test scores, post-test differences favored the exercise group in shoulder position (p = 0.001) and performance (p = 0.02). Training also reduced dominant shoulder pain, reinforcing between-group differences at post-test. |
| Khakpour-fard et al., (2023) Iran | RCT | N = 30 Sex = 30 males Age = 26.8 ± 5.5 years SD type = NR | D = 8 weeks F = 3 per week I = NR T = 25–30 minutes T = suspension training | NR | Internal and external rotator muscle strength, functional stability, and proprioception | There were significant time-by-group interactions for internal rotator strength, external rotator strength, functional stability, and shoulder proprioception accuracy (p = 0.001), indicating that changes over time differed between groups. Additionally, there were significant main effects of time and training across all variables. Specifically, internal rotator strength showed significant effects of time (p = 0.005) and training (p = 0.021); external rotator strength showed time (p = 0.003) and training (p = 0.009); functional stability improved with both time (p = 0.001) and training (p = 0.001); and proprioception accuracy improved with time (p = 0.001) and training (p = 0.001). |
| Paraskev-opoulos et al., (2022) Greece | RCT | N = 39 Sex = NR Age = 21.8 ± (NR) years SD type = NR | D = 6 weeks F = 3 per week I = NR T = 60 minutes T = EG1: kinetic chain approach; EG2: mirror cross exercise | NR | Functional throwing performance index Throwing performance (velocity, strength) | The Functional throwing performance Index and throwing velocity significantly improved in both the kinetic chain approach (p < 0.011 and p = 0.001) and mirror cross exercise (p = 0.004 and p < 0.001) groups, with no changes in controls. Throwing force increased significantly only in the mirror cross exercise group (P = 0.011). |
| Naderifar and Ghan-bari, (2022) Iran | Non-RCT | N = 54 Sex = 54 females Age = 22.2 ± 2.4 years SD type = NR | D = 8 weeks F = 3 per week I = moderate T = NR T = Selected Corrective Exercises | Typical training regimen | Internal and external rotation ROM | Results revealed that, in the experimental group, glenohumeral internal rotation significantly increased (p = 0.001) following the exercise program. No significant changes were observed in the control group. |

*(Continued)*

| Study details | Study design | Sample description | Exercise characteristics of EG | CG intervention | Sports performance measures | Main outcomes |
|---|---|---|---|---|---|---|
| Song et al., (2020) Republic of Korea | Non-RCT | N = 27 Sex = 27 males Age = 19.6 ± 1.9 years SD type = Sick | D = 8 weeks F = 3 per week I = NR T = 40 minutes T = Scapular Kinetic Chain Exercise | NR | Muscle activation | Maximal and mean muscular activation significantly increased after exercise in Normal-Dominant and SICK-Dominant upper and lower trapezius muscles (p < 0.05). The SICK-Dominant serratus anterior showed lower activation than Normal-Dominant at pre-test (p = 0.034), with differences persisting post-test compared to Normal-Non-Dominant (p = 0.031) |
| Sant et al., (2018) United Kingdom | RCT | N = 25 Sex = 25 males Age = 23.2 ± 3.6 years SD type = Unilateral | D = NR F = NR I = NR T = NR T = Prehabilitation | Usual routine | Functional throwing performance index, power, upper extremity stability | Pain was reported in 3 athletes in the control group versus 1 in the study group (p = 0.59). Athletes receiving prehabilitation showed significantly greater improvements in external rotation (p = 0.01) and internal rotation (p = 0.03) compared to controls. No significant differences were found between groups in functional tests, scores, or abduction strength. |
| Ilyoung et al., (2018) Republic of Korea | RCT | N = 24 Sex = 24 males Age = 25.7 ± 1.4 years SD type = Inferomedial winging and medial border winging | D = 6 weeks F = 3 per week I = 70%−90% stretch T = NR T = EG1: PSSE group; EG2: SSE | NR | Isokinetic peak moment/body weight, ROM, Pain | Significant time × group interactions were found for concentric and eccentric external rotation peak moment/body weight (p = 0.039, p = 0.008), ERe to IRc ratio (p = 0.025), and rotation ROM (IRROM p < 0.001, ERROM p = 0.001). The PSSE group showed improvements at 6 weeks in ERc, ERe, ERe/IRc ratio, IRROM (↑15°), ER ROM (↓12°), and GIRD (↓17°); the SSE group did not show significant changes in strength or ROM. Pain decreased over time in both groups (p < 0.001) with no group interaction (p = 0.56). |
| Nowotny et al., (2018) Germany | RCT | N = 28 Sex = 16 males and 12 females Age = 33 ± (NR) years SD type = type I | D = 6 weeks F = 2 per week I = NR T = 60 minutes T = specific exercise | Massage Therapy | ROM, Pain, disability, Scapular kinematics, shoulder function | Both exercise and massage reduced pain (VAS: exercise p = 0.007; control p = 0.004), but only the exercise group showed significant improvement in shoulder function (QuickDASH p = 0.001; SICK Scapula p = 0.003; Hand Press-up p = 0.026). |
| Moura et al., (2016) Brazil | Non-RCT | N = 4 Sex = 2 males and 2 females Age = 24.7 ± (NR) years type = NR | D = 1 session F = NR I = NR T = 120 minutes T = Specific training | NR | ROM, pain, sports performance, muscle activation, strength, function | Participants showed reduced pain, improved function and performance, increased shoulder strength, greater ROM, and enhanced serratus anterior activation. |
| Merolla et al., (2010) (a) Italy | Non-RCT | N = 29 Sex = 18 males and 11 females Age = 23 ± 4.2 years SD type = NR | D = 24 weeks F = NR I = NR T = NR T = Rehabilitation program for restoring scapular muscular control and balance | NR | Strength, pain, ROM | Muscle strength of the supraspinatus and infraspinatus, measured by EC and IST tests, significantly increased at 3- and 6-months post-rehabilitation (p < 0.01). Additionally, glenohumeral internal rotation ROM improved at both time points. Patient pain intensity decreased significantly from 7.5 ± 2.3 at baseline to 3.4 ± 1.8 at 3 months and 2.9 ± 2.1 at 6 months (p < 0.01). |

*(Continued)*

**Table 3.** (Continued)

| Study details | Study design | Sample description | Exercise characteristics of EG | CG intervention | Sports performance measures | Main outcomes |
|---|---|---|---|---|---|---|
| Merolla et al., (2010) (b) Italy | Non-RCT | N = 29 Sex = 16 males and 13 females Age = 23 ± 4.5 years SD type = NR | D = 24 weeks F = NR I = NR T = NR T = Rehabilitation program for restoring scapular muscular control and balance | NR | ROM, strength | Isometric strength of the infraspinatus muscle, assessed using the infraspinatus strength test, significantly increased after 6 months—3.3 ± 1.54 kg for examiner 1 (p = 0.0069) and 3.9 ± 1.6 kg for examiner 2 (p = 0.0058). The mean difference between infraspinatus strength test and the infraspinatus scapular retraction test results at 6 months was not statistically significant (p = 0.061). Glenohumeral internal rotation also showed significant improvement, increasing from 54.5 ± 9.8 to 67.3 ± 10.1 degrees for examiner 1 (p = 0.0096) and from 53.9 ± 10.2 to 68.1 ± 11.4 degrees for examiner 2 (p = 0.0089) |
| Merolla et al., (2010) (c) Italy | Non-RCT | N = 31 Sex = 22 males and 9 females Age = 22 ± 2.5 years SD type = NR | D = 24 weeks F = NR I = NR T = NR T = Rehabilitation program for restoring scapular muscular control and balance | NR | Pain, strength | The mean force values of the infraspinatus strength test increased significantly after 3 months (p < 0.01) and 6 months (p < 0.001) of rehabilitation. The mean difference between infraspinatus strength test and the infraspinatus scapular retraction test decreased from 4.72 ± 0.007 at baseline to 1.2 ± 0.26 at 3 months and 0.4 ± 0.006 at 6 months. Mean pain scores were 2.4 ± 1.8 at 3 months and 2.6 ± 1.4 at 6 months. |

**Descriptions:** N: number of participants, F: frequency, I: intensity, D: duration, T: time, IG: intervention group, CG: control group, NR: not applicable, RCT: randomized controlled trial, SD: scapular dyskinesis, SD type not reported (NR) in some studies diagnosis based on clinical or visual assessment, without standardized criteria ROM: range of motion, IRROM: Internal Rotation Range of Motion, ERROM: External Rotation Range of Motion, SDBET: scapular dyskinesis-based exercise therapy, PSSE: posterior shoulder stretch combined with a scapular stabilization exercise, SSE: scapular stabilization exercise without stretching, MPT: multimodal physical therapy. Merolla et al. (2010 a/b/c) report outcomes from subgroups of the same study. Sample sizes, sex distribution, and mean ages differ slightly across subgroups.

corrective interventions [47]. Similarly, Nowotny et al. (2018) reported a marked decrease in the prevalence of scapular dyskinesis after a training period [42].

**3.3.5 Pain and disability.** Several studies have reported significant reductions in pain and disability. In an 8-week program with three sessions per week, Wen et al. (2025) observed pain reduction in both the Scapular Dyskinesis–Based Exercise Therapy group and the Multimodal Physical Therapy group; however, only the Dyskinesis–Based Exercise Therapy maintained this improvement at the 12-week follow-up, indicating greater long-term effectiveness [7]. Similarly, Karimi and Firouzjah (2024) reported significant reductions in dominant-shoulder pain compared to the control group following an 8-week scapular stabilization exercise program (40 minutes per session) [20]. Khakpourfard et al. (2023) also demonstrated pain reduction after an 8-week suspension-based exercise program with shorter session durations (25–30 minutes) [18]. Moura et al. (2016) reported pain reduction in all participants after a 6-week rehabilitation program with only two sessions per week, despite the lower session frequency [44]. In contrast, Sant et al. (2018) observed a reduction in pain following pre-treatment, but this change was not statistically significant, possibly due to insufficient treatment intensity or duration [41]. In higher-intensity protocols, Ilyoung et al. (2018), compared two 6-week programs with seven sessions per week and stretching intensities of 70–90%, reporting significant pain reductions in both groups but no meaningful differences between them, suggesting that while higher intensity can facilitate pain reduction, the specific exercise modality may be less critical [45]. Finally, Merolla et al. (2010) reported a significant reduction in pain

at both 3- and 6-month follow-ups following a 24-week rehabilitation program, demonstrating sustained long-term effects [23,25]. Nowotny et al. (2018) reported a significant reduction in pain levels on the visual analog scale in both the exercise (p = 0.007) and control (p = 0.004) groups, while a significant improvement in QuickDASH scores was observed only in the exercise group (p = 0.001) [42]. Overall, 6- to 8-week programs with 2–3 sessions per week typically yield significant pain reduction; however, programs with longer follow-up periods or optimized training intensity tend to produce more durable, clinically meaningful outcomes. Table 4 summarizes the changes in pain and sports performance parameters resulting from exercise-based interventions.

### 3.4  Quality assessment

Among the included studies in this review, 8 were RCTs evaluated using the RoB-2 tool [7,18–20,41–43,45] (Fig 2), while the remaining six non-RCTs were assessed with the ROBINS-I tool [23–25,44,46,47] (Fig 3). According to the overall RoB-2 assessment, five of the eight randomized controlled trials (62.5%) were rated as having a low risk of bias [7,18,41,43,45]. whereas three studies (37.5%) were classified as having some concerns [19,20,42]. Regarding bias arising from the randomization process, four studies (50%) were rated as low risk [7,19,43,45]. while four studies (50%) showed some concerns [18,20,41,42]. Regarding Domain 2 (Bias due to deviations from intended interventions), four studies (50%) were rated as low risk [7,18,43,45], while four studies (50%) showed some concerns [19,20,41,42]. For bias due to missing data, seven studies (87.5%) were rated as low risk [7,18,20,41–43,45], and one study (12.5%) was rated as some concerns [19]. In the domain of bias in the measurement of outcomes, five studies (62.5%) demonstrated a Low risk (6, 16, 38–40), whereas three studies (37.5%) had some concerns [19,20,45]. Finally, regarding bias in the selection of the reported results, all eight RCTs (100%) were rated as having a low risk. [7,18–20,41–43,45]. The six non-RCTs, assessed with ROBINS-I, demonstrated varied risk profiles. In Domain 1 (bias due to confounding), 57.1% of studies were rated moderate risk, [23–25,46], 14.3% low risk [47], and 14.3% high risk [44], highlighting significant challenges in controlling confounding variables and emphasizing the need for cautious interpretation of the results All studies (100%) were rated low risk in Domain 2 (bias in selection of participants) and Domain 3 (bias in classification of interventions) [23–25,44,46,47]. reflecting rigorous participant selection and accurate classification. Domain 4 (bias due to deviations from intended interventions) was also low risk across all studies (100%), indicating strong adherence to study protocols [23–25,44,46,47]. In Domain 5 (bias due to missing data), all studies were rated as low risk [23–25,44,46,47]. suggesting effective data management in most cases. Domain 6 (bias in measurement of outcomes) had 83.3% of studies at moderate risk [23–25,46,47], and 14.3% at high risk [44]. indicating potential concerns regarding measurement accuracy. Finally, Domain 7 (bias in selection of the reported result) was low risk for all studies (100%), demonstrating transparent and comprehensive reporting practices [23–25,44,46,47]. Inter-rater agreement was perfect (κ = 1.0), confirming consistent bias assessments across reviewers.

**Table 4. Changes in pain and sports performance parameters influenced by exercise-based interventions.**

| Parameters | Number of studies | Significant negative effect | Significant positive effect | No significant effect |
|---|---|---|---|---|
| Upper limb function and performance | 4/14 studies | NA | [18,20,41–43] | NA |
| Range of motion | 6/14 studies | NA | [7,23,24,44,45,47] | NA |
| Muscle activity and strength | 8/14 studies | NA | [18,23–25,44–46] | NA |
| Scapular kinematics | 4/14 studies | NA | [7,19,42,47] | NA |
| Pain and disability | 8/14 studies | [7,18,20,42,44,45] | | [41] |

**Descriptions:** NA: not applicable, No significant effect reported for these outcomes in the included studies; only studies reporting data are reflected in the table.

**A**

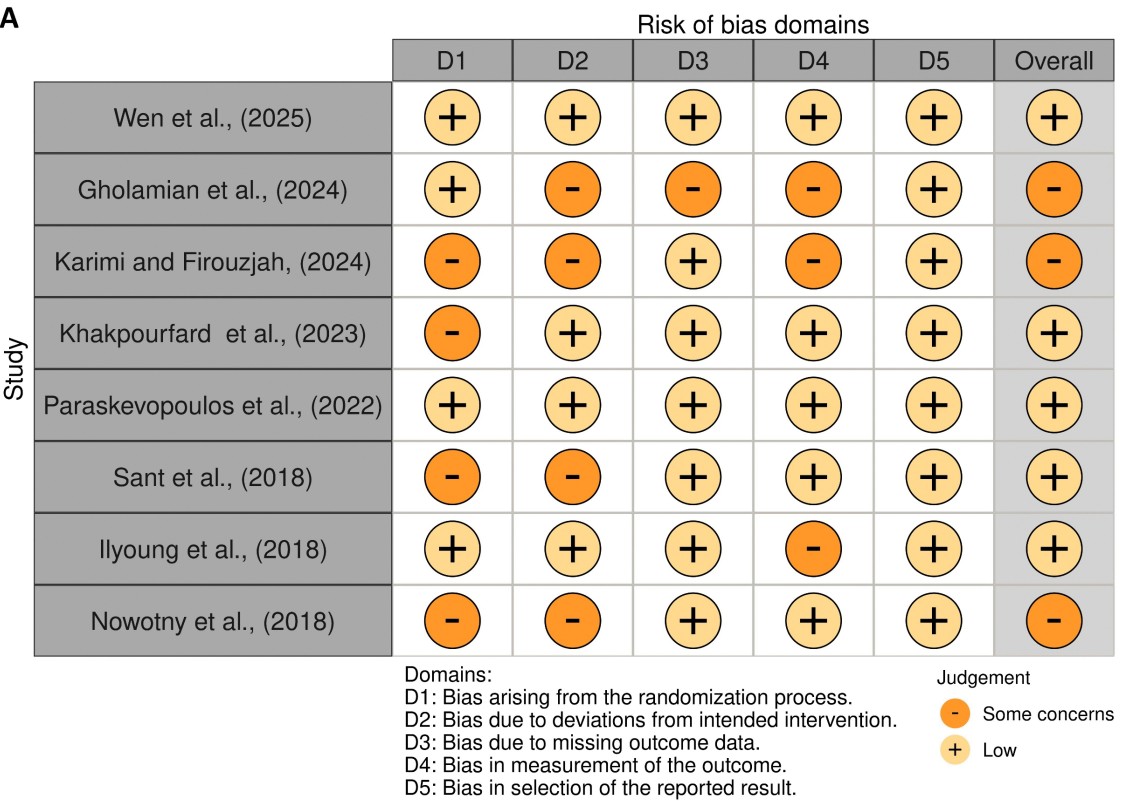

Domains:
D1: Bias arising from the randomization process.
D2: Bias due to deviations from intended intervention.
D3: Bias due to missing outcome data.
D4: Bias in measurement of the outcome.
D5: Bias in selection of the reported result.

Judgement
- Some concerns
+ Low

**B**

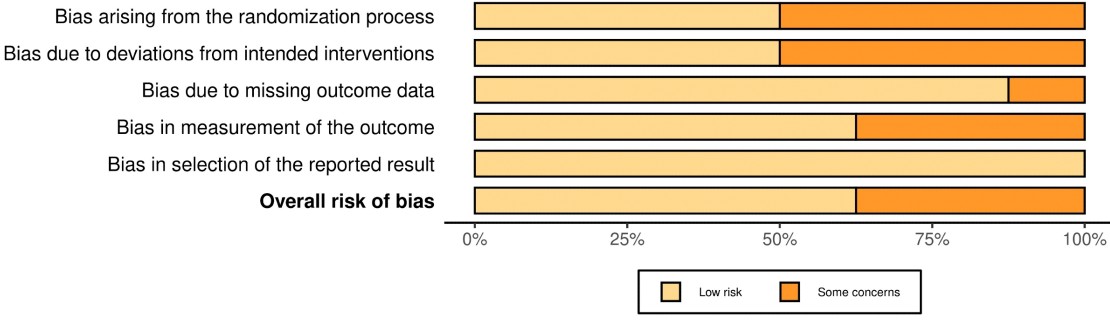

**Fig 2. Summary of risk of bias among the included RCTs assessed via ROB-2.** A: Traffic light plot; B: Summary plot.

### 3.5 Certainty of the evidence

Using GRADE, certainty ranged from moderate to very low. Evidence was moderate for improvements in shoulder function/disability, and in glenohumeral ROM; low for pain, rotator cuff/scapular strength, scapular kinematics/rhythm, and muscle activation; and very low for sport-specific performance. Downgrades were primarily driven by imprecision (small sample sizes and lack of pooled estimates) and inconsistency (heterogeneous interventions, comparators, and outcome measures). The risk of bias was frequently unclear due to limited blinding, and publication bias could not be assessed. A summary of the findings is provided in Table 5.

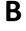

**A**

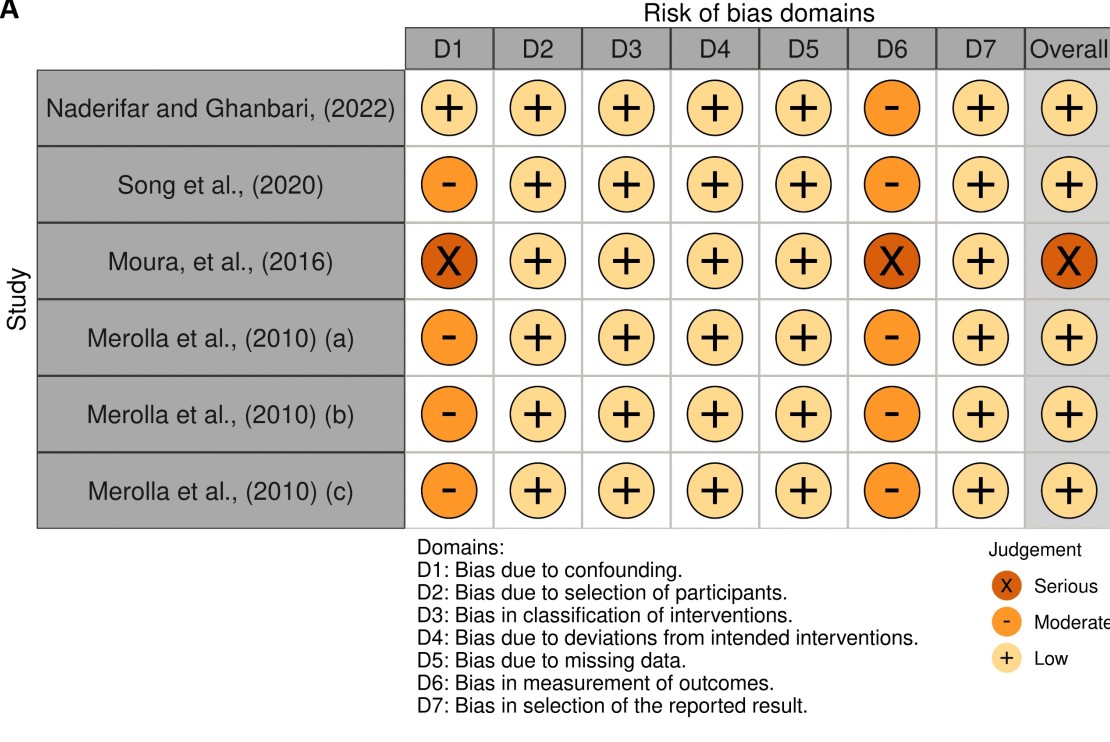

Domains:
D1: Bias due to confounding.
D2: Bias due to selection of participants.
D3: Bias in classification of interventions.
D4: Bias due to deviations from intended interventions.
D5: Bias due to missing data.
D6: Bias in measurement of outcomes.
D7: Bias in selection of the reported result.

Judgement

X Serious
- Moderate
+ Low

**B**

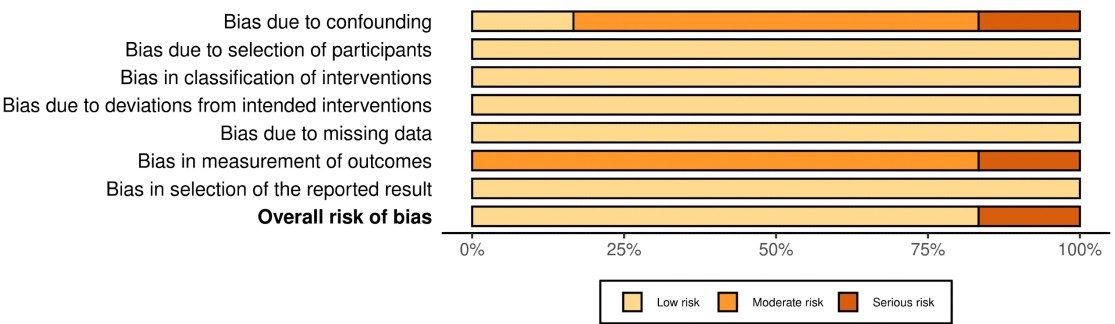

**Fig 3. Summary of risk of bias among the included non-RCTs assessed via ROBINS-I.** A: Traffic light plot; B: Summary plot.

## 4. Discussion

This review underscores the effectiveness of movement-based exercise programs for athletes with scapular dyskinesis. These programs are likely to enhance shoulder function and increase glenohumeral ROM, with moderate certainty. However, the impact on pain and strength is inconsistent and appears to be protocol-dependent. Some trials indicate improvements in scapular kinematics, though the evidence surrounding this is of low certainty. Critically, only two of the included studies measured true sport-specific performance metrics. Therefore, while clinical improvements in function and ROM are evident, any extrapolation of these findings to direct enhancements in athletic performance (e.g., velocity, power) remains speculative due to this limited and insufficient evidence base. Overall, incorporating scapular-focused exercises in rehabilitation appears beneficial for improving shoulder function, ROM, and pain; however, due to the very low certainty

**Table 5. Summary of findings (GRADE): Exercise-based interventions for scapular dyskinesis (vs. control/usual training or alternative therapy).**

| Outcome | Assessed with | Partic-ipants, design | Effect | Certainty (GRADE) | Reasons for rating/key comments |
|---|---|---|---|---|---|
| Pain | VAS/NRS | n = 109, 4 RCTs | Direction of effect favors exer-cise overall; between-group differences inconsistent (some improvements in both arms; one trial favored comparator at 8 wks, no difference at 12 wks). | Low ●●○○ | ↓ for inconsistency (heterogeneous directions/magnitudes) and imprecision (small samples; no pooled estimate). |
| Shoulder function/ disability | QuickDASH/ disability index/ upper-limb function scores | n = 115, 4 RCTs | Probably improved function/dis-ability vs control/alternative; one trial showed no between-group difference on functional tests. | Moderate ●●●○ | ↓ for imprecision (small total N, wide CIs encompassing both benefit and no effect). No downgrade for inconsis-tency: variability in effect size is explainable by differences in intervention content (e.g., exercise specificity) and choice of functional outcome measure. Risk of bias was not a common reason for downgrade across these studies. |
| Gleno-humeral ROM | Goniometry/ isokinetic ROM | n = 81, 3 RCTs | Generally positive effects of exer-cise on IR ROM vs control/alter-native. Effects on ER ROM were less consistent, with one study showing a decrease concomitant with IR increase [44]. | Moderate ●●●○ | ↓ for imprecision (small sample sizes, CIs not reported/ calculable for all comparisons). No downgrade for incon-sistency: the direction of effect for the primary ROM deficit (IR) was consistent; variability in ER ROM is a biologically plausible, protocol-dependent adaptation not considered contradictory. |
| Shoulder strength | Hand-held dynamometry/ isokinetic peak torque | n = 101, 3 RCTs | Mixed: several trials report gains in ER/IR or force; others show gains only in a specific program. | Low ●●○○ | ↓ for inconsistency (program-dependent effects) and imprecision (small N; no pooled CI). |
| Scapular kinematics/ rhythm | Scapular kinematics; scapulohu-meral rhythm | n = 62, 2 RCTs | Improvements in rhythm/kinemat-ics during exercise are reported in exercise groups; the magnitude and responder proportions vary by protocol. | Low ●●○○ | ↓ for imprecision (very small evidence base) and incon-sistency (heterogeneous definitions/measures; partial responders). |
| Sports per-formance | Functional Throwing Performance Index: ball velocity/force | n = 64, 2 RCTs | Conflicting: one RCT shows meaningful gains in throwing index/velocity; another shows no between-group differences on functional tests. | Very low ●○○○ | ↓ for risk of bias (performance/measurement blinding unlikely), inconsistency (opposite findings), and impreci-sion (small total N; no pooled CI). |

**Abbreviations and descriptions:** CI = confidence interval; ER = external rotation; GRADE = Grading of Recommendations, Assessment, Develop-ment and Evaluation; IR = internal rotation; n = number of participants; Non-RCT = non-randomized controlled trial; NRS = Numeric Rating Scale (pain); PT = physical therapy; QuickDASH = Quick Disabilities of the Arm, Shoulder and Hand questionnaire; RCT = randomized controlled trial; ROM = range of motion; SDBET = scapular dyskinesis–based exercise therapy; VAS = Visual Analog Scale (pain). Certainty ratings: ●●●● High, ●●●○ Moderate, ●●○○ Low, ●●○○○ Very low.

of the evidence regarding sport-specific performance, no firm recommendations can be made for performance outcomes, and further standardized, adequately powered trials are needed.

## 4.1 Impact on pain and disability

Pain reduction was often observed after engaging in movement-based exercises, which aligns with the objective of addressing altered biomechanics and compensatory movement patterns. However, the effectiveness of these exercises relative to other treatments was inconsistent across trials. Athletes with scapular dyskinesis often experience pain, which is commonly associated with altered biomechanics and compensatory movement patterns that may place excessive

strain on the shoulder complex [2,29]. The studies included in this review consistently reported that athletes experienced decreased pain levels following structured exercise programs, particularly those focusing on scapular stabilization and neuromuscular control [7,20,23,25,42,44,45]. For instance, Wen et al. (2025) demonstrated that participants in the Scapular Dyskinesis-Based Exercise Therapy group-maintained pain reduction at the 12-week follow-up, indicating the long-term benefits of such interventions [7]. This sustained improvement in pain levels suggests that movement-based exercises not only provide immediate relief but also contribute to lasting changes in shoulder function and mechanics. The ability to maintain reduced pain levels over time is crucial for athletes, as it allows them to train and compete without fear of exacerbating their condition [48]. The mechanisms underlying pain reduction in athletes undergoing movement-based exercises may be multifaceted [49–51]. First, these exercises often emphasize proper scapular positioning and movement patterns, which can help restore normal biomechanics and reduce the strain on the shoulder joint [7]. By improving scapular stability and coordination, athletes may experience less discomfort during overhead activities, which could be associated with enhanced performance and reduced disability [29]. Additionally, incorporating neuromuscular training may enhance proprioception and motor control, further contributing to pain alleviation by promoting more efficient movement strategies [52,53]. Moreover, the psychological aspects of pain management should not be overlooked. Engaging in a structured exercise program can empower athletes, providing them with a sense of control over their recovery process. This empowerment can lead to improved mental well-being and reduced pain perception, as athletes may feel more confident in their ability to manage their condition. The positive feedback loop created by pain reduction and improved function can significantly enhance an athlete's overall quality of life [54,55].

This aligns with previous literature suggesting that targeted rehabilitation can effectively alleviate pain and improve quality of life in individuals with shoulder dysfunction [56,57]. Studies have shown that exercise interventions focusing on scapular dyskinesis can lead to significant improvements in pain intensity [42,45]. In summary, this review's evidence underscores the effectiveness of movement-based exercises in reducing pain and disability among athletes with scapular dyskinesis. By addressing the underlying biomechanical issues and promoting proper movement patterns, these interventions not only alleviate pain but also enhance overall shoulder function, allowing athletes to return to their sport with greater confidence and reduced risk of re-injury.

## 4.2 Improvements in ROM and muscle strength

Movement-based exercises were linked to improved shoulder ROM, particularly in internal rotation, which supports the demands of overhead sports (moderate certainty). Enhanced ROM is critical for athletes, particularly in overhead sports, where shoulder mobility is essential for optimal performance [58,59]. Limitations in shoulder ROM can significantly hinder an athlete's ability to execute skills effectively, such as throwing, serving, or spiking, which require a full ROM for maximal power and precision [60–62]. Studies such as those by Naderifar and Ghanbari (2022) and Khakpourfard et al. (2023) reported significant gains in both internal and external rotation [18,47]. For instance, Naderifar and Ghanbari (2022) demonstrated that participants in their study experienced marked improvements in glenohumeral internal rotation following a structured exercise program. This increase in internal rotation is particularly important for athletes involved in sports that require overhead motions, as it allows for more effective arm positioning and force generation during performance [18,47]. Furthermore, increased muscle strength, particularly in the rotator cuff and scapular stabilizers, supports the notion that stabilization exercises can restore normal scapular mechanics and improve overall shoulder function [63,64]. The rotator cuff muscles play a crucial role in stabilizing the glenohumeral joint during dynamic movements, and their strength is essential for maintaining proper shoulder mechanics [65,66]. Studies have shown that various interventions can lead to significant improvements in the strength of these muscles, which in turn enhance shoulder joint stability and reduce the risk of injury [67–69]. Additionally, the improvements in muscle strength observed in the included studies were not limited to the rotator cuff; athletes also demonstrated enhanced strength in the scapular stabilizers, such as the serratus anterior and trapezius muscles [70]. These muscles are crucial for maintaining proper scapular positioning during arm movements,

and their strengthening can improve scapular kinematics [70,71]. As a result, athletes may experience not only enhanced performance but also a reduced likelihood of developing shoulder-related injuries [72].

In summary, this review's evidence underscores the importance of movement-based exercises for improving ROM and muscle strength in athletes with scapular dyskinesis. By addressing these critical components, rehabilitation programs can facilitate improved shoulder function, ultimately contributing to enhanced athletic performance and a reduced risk of injury.

### 4.3 Scapular kinematics and functionality

Findings on scapular kinematics suggest exercise may enhance scapular control, underscoring its importance for shoulder function [73]. However, heterogeneous measures and small samples temper confidence in these effects and their translation to sport performance (low certainty). Several studies indicated that movement-based exercises led to improved scapulohumeral rhythm and reduced scapular dyskinesis [19,43]. Scapulohumeral rhythm refers to the coordinated movement of the scapula and humerus during shoulder motion, which is essential for maintaining optimal shoulder mechanics [74]. Disruptions in this rhythm may be associated with compensatory movement patterns, increased strain on the shoulder joint, and a higher likelihood of injury [75]. For instance, Gholamian et al. (2024) reported significant enhancements in scapulohumeral rhythm following a functional training program, indicating that structured exercise can facilitate the restoration of normal scapular mechanics [19]. In summary, the evidence emphasizes the role of movement-based exercises in enhancing scapular kinematics and function. By improving scapulohumeral rhythm and reducing dyskinesis, these interventions can foster better neuromuscular control, optimize shoulder mechanics, and ultimately decrease injury risk for athletes. This underscores the importance of incorporating targeted rehabilitation strategies into training regimens for athletes with scapular dyskinesis.

### 4.4 Limitations and future directions

Despite the promising findings, this review has several limitations. Heterogeneity in sample sizes, intervention protocols, and outcome measures across the included studies limits the ability to draw definitive conclusions. The small number of studies and heterogeneity in designs, populations, and interventions prevented subgroup or sensitivity analyses, which limits the interpretability and generalizability of the findings. According to the GRADE assessment, the certainty of the evidence was moderate for shoulder function and disability, as well as ROM, and low for pain and strength, and very low for sport-specific performance. Most significantly, only two studies assessed true sport-specific outcomes, fundamentally limiting conclusions in this area. These ratings were primarily driven by small overall sample sizes, imprecision, and inconsistencies among studies. Publication bias could not be formally assessed because the number of included studies per outcome was insufficient to generate funnel plots, potentially affecting the certainty of the findings. The lack of reported exercise intensity limits the clinical reproducibility and generalisability of the findings. To enhance the generalisability of the results, future research should use standardised intervention protocols and larger sample sizes. Most critically, future trials must prioritize the inclusion of objective, sport-specific performance metrics as primary endpoints to directly investigate the link between dyskinesis correction and athletic performance. Additionally, longitudinal studies are warranted to clarify the long-term effects of movement-based exercises on sports performance and injury prevention.

### 4.5 Clinical implications

The implications of this review are significant for clinicians and sports professionals. The evidence supports integrating movement-based exercises into rehabilitation programs for athletes with scapular dyskinesis to improve shoulder function, ROM, and kinematics. However, clinicians should be cautious about directly extrapolating these clinical improvements to guaranteed gains in sport-specific performance, as this link remains unproven. By focusing on improving scapular stability, strength, and coordination, practitioners can enhance shoulder health and reduce the likelihood of shoulder injuries.

Furthermore, the findings emphasize the need for individualized rehabilitation programs that account for each athlete's specific needs and characteristics.

## 5. Conclusion

Movement-based rehabilitation for athletes with scapular dyskinesis is associated with probable improvements in shoulder function/disability, and in glenohumeral ROM, over 6–12 weeks (moderate certainty). The effects on pain and strength are uncertain (low certainty), and the impact on sport-specific performance remains highly uncertain (very low certainty) due to small, heterogeneous trials and imprecision. Scapular kinematics may improve with targeted protocols, but measurement variability limits confidence (low certainty). Given the very low certainty of evidence for sport-specific performance, no clinical recommendation can be made regarding performance outcomes. Rigorous, adequately powered RCTs using standardized protocols, core outcome sets, including sport-specific metrics, and longer follow-up are needed to confirm benefits and clarify effects on performance and injury risk.

## Supporting information

**S1 File. PRISMA checklist.**
(DOCX)

**S2 File. Reasons for exclusion of studies.**
(DOCX)

## Author contributions

**Conceptualization:** Hossein Khazanin, Hassan Daneshmandi, Mohammad Alghosi.

**Data curation:** Narges Norouzi, Yaser Dehghan, Bahareh Sadegh, Hossein Khazanin, Mohammad Alimoradi, Hassan Daneshmandi, Mohammad Alghosi.

**Formal analysis:** Hossein Khazanin, Mohammad Alghosi.

**Methodology:** Mònica Solana–Tramunt, Narges Norouzi, Bahareh Sadegh, Hossein Khazanin, Mohammad Alimoradi, Mohammad Alghosi.

**Project administration:** Mohammad Alghosi.

**Software:** Mohammad Alghosi.

**Supervision:** Hassan Daneshmandi, Mohammad Alghosi.

**Visualization:** Mohammad Alghosi.

**Writing – original draft:** Hossein Fakoor Rashid, Narges Norouzi, Yaser Dehghan, Bahareh Sadegh, Hossein Khazanin, Mohammad Alimoradi, Mohammad Alghosi.

**Writing – review & editing:** Mònica Solana–Tramunt, Hossein Fakoor Rashid, Hassan Daneshmandi, Mohammad Alghosi.

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
