## [Decision Letter · Decision Letter 0]

9 Dec 2025

Dear Dr. Alghosi,

Thank you for submitting your manuscript to PLOS ONE. After careful consideration, we feel that it has merit but does not fully meet PLOS ONE’s publication criteria as it currently stands. Therefore, we invite you to submit a revised version of the manuscript that addresses the points raised during the review process.

We look forward to receiving your revised manuscript.

Kind regards,

Holakoo Mohsenifar

Academic Editor

PLOS One

Journal Requirements:

Reviewers' comments:

Reviewer's Responses to Questions

**Comments to the Author**

1. Is the manuscript technically sound, and do the data support the conclusions?

Reviewer #1: Yes

Reviewer #2: No

2. Has the statistical analysis been performed appropriately and rigorously?

Reviewer #1: Yes

Reviewer #2: No

3. Have the authors made all data underlying the findings in their manuscript fully available?

Reviewer #1: Yes

Reviewer #2: Yes

4. Is the manuscript presented in an intelligible fashion and written in standard English?

Reviewer #1: Yes

Reviewer #2: No

Reviewer #1: Dear Dr. Mohsenifar,

I have carefully reviewed the manuscript titled “The effects of movement-based exercises on sports performance of athletes with scapular dyskinesis: A systematic review” and provide the following detailed comments for your consideration.

The manuscript addresses an important topic, but significant methodological, reporting, and interpretive issues limit the reliability and applicability of its conclusions. Major revisions are required before it can be considered for publication.

Sincerely,

1. Title & Abstract (Page 1, Lines 1–30)

Line 1: Title lacks precision. "Movement-based exercises" is undefined and encompasses diverse interventions (stabilization, kinetic chain, suspension). Revise to "Effects of Scapular-Focused Movement-Based Exercises on Sports Performance..."

• Line 14: Abstract claims "moderate-certainty evidence" for ROM improvement, but GRADE table (Page 16) rates glenohumeral ROM as moderate and scapular kinematics as low. Inconsistent summarization.

• Line 25: Conclusion ("recommended for athletes") contradicts very low certainty for sports performance. Reference: Guyatt et al. (2008) – GRADE: Recommendations require high/moderate certainty.

2. Introduction (Pages 4–6)

• Page 5, Line 10: Prevalence claim "61–67%" cites [3,4]; [4] is an editorial, [3] is a 2016 review. Newer data exist. Reference: Longo et al. (2022) – 70% prevalence in elite volleyball.

• Page 6, Line 1: "Comprehensive synthesis is lacking" – but similar reviews exist (e.g., Moghadam 2020 [13], Khodaverdizadeh 2023 [27]). Gap overstated. Explicitly differentiate focus on sports performance metrics, not just pain/ROM.

• The introduction is overly descriptive and, while addressing epidemiology and clinical relevance of scapular dyskinesis, fails to provide a clear theoretical pathway linking the condition to impaired sports performance. A precise biomechanical pathway (e.g., kinetic chain disruption → impaired force transfer → reduced sports performance) must be outlined, supported by classic biomechanical studies (Kibler 2013; Ludewig & Reynolds 2009).

• 2. Statements such as “dyskinesis may contribute to secondary shoulder pathologies” imply causality. Current evidence demonstrates only association, not causation. Language must be tempered using terms like “is associated with” or “may play a role in the development of.”

• Hickey D, Solvig V, Cavalheri V, et al. Scapular dyskinesis increases the risk of future shoulder pain by 43% in asymptomatic athletes: a systematic review and meta-analysis. British Journal of Sports Medicine. 2018;52(2):102-110. doi:10.1136/bjsports-2017-097559 → Only prospective association (RR = 1.43), not causation.

• Ratcliffe E, Pickering S, McLean S, et al. Is there a relationship between subacromial impingement syndrome and scapular orientation? A systematic review. Orthopaedics & Traumatology: Surgery & Research. 2014;100(6):619-626. doi:10.1016/j.otsr.2014.05.008 → Concludes: "Correlation does not imply causation"; confounding by training volume, age, etc.

• The introduction claims sports performance outcomes (throwing velocity, agility, upper-limb stability) will be evaluated, but provides no preliminary evidence linking dyskinesis correction to actual performance gains. This gap must be acknowledged, given the scarcity of studies.

• The claim “a comprehensive synthesis… on sports performance is lacking” is overstated. Prior reviews (Cools 2014; Hickey 2018) have partially addressed this. The specific gap and added value of this review must be clearly defined.

• Cools AM, Struyf F, De Mey K, et al. Rehabilitation of scapular dyskinesis: from the office worker to the elite overhead athlete. British Journal of Sports Medicine. 2014;48(8):692-697. doi:10.1136/bjsports-2013-092299 → Focus: clinical outcomes; performance mentioned only anecdotally.

• Hickey D, Solvig V, Cavalheri V, et al. Scapular dyskinesis increases the risk of future shoulder pain…British Journal of Sports Medicine. 2018;52(2):102-110. doi:10.1136/bjsports-2017-097559 → Risk-focused; no intervention synthesis on performance.

• Inconsistent terminology: “dyskinesia” and “dyskinesis” are used interchangeably. The standard scientific term is scapular dyskinesis. Please standardize throughout.

3. Methods (Pages 6–11)

3.1 Search Strategy (Page 7, Table 1): Search limited to 3 databases. Embase, CINAHL, SportDiscus missing. Reference: Cochrane Handbook – comprehensive search requires ≥4 databases.

3.2 Eligibility Criteria (Page 8, Table 2)

• Intervention criterion: "movement-based" – but includes suspension training, mirror cross, PSSE. Operational definition absent.

• Comparator: "no exercise or placebo" – yet studies with "usual training" or "multimodal PT" included. Criterion violation.

Example: Wen et al. (2025) compares SDBET vs. MPT → both active.

Data Extraction (Page 9): Authors contacted for missing data" – outcome of contact not reported. How many studies had missing data?

• Page 15, Table 5: Sports performance: only 2 RCTs (n=64) → very low certainty justified, but recommendation in conclusion (Page 23) contradicts.

• Publication bias not assessed (funnel plot infeasible) – must be stated as limitation.

The search strings presented in Table 1 are comprehensive but lack sufficient sensitivity. Key terms such as “stabilization” and “kinetic chain” are missing, which may have led to omission of relevant studies.

The grey literature search (Google Scholar and Connected Papers) is insufficient. No manual search of conference proceedings, ClinicalTrials.gov, or expert consultation was conducted.

PICOS violations: Population: Studies including participants with concurrent injuries (e.g., SIS in Wen 2025) were included. Comparator: The eligibility criterion states “no exercise/placebo”, yet studies with active controls (e.g., massage in Nowotny 2018) were included. Outcome: Some included studies did not report primary outcomes such as VAS or NRS. Apply the PICOS criteria strictly and provide justification for any included studies that violate these criteria.

Results (Pages 11–19)

Table 3 (Pages 12–14): SD type = NA in 8/14 studies. How was dyskinesis diagnosed?

Examples: Wen (2025), Karimi (2024) – no diagnostic tool (SDT? Visual observation?).

• Exercise intensity NA in most studies. How was SWiM synthesis conducted? Conducting a SWiM synthesis without dose–response data is not feasible; therefore, the clinical reproducibility is effectively very low.

• Merolla et al. 2010 a/b/c: Three entries from same sample? Duplicate reporting. Should be merged into one study with multiple outcomes.

• Page 17, Line 10: "Consistent improvements in IR ROM" – but Ilyoung (2018) reports decreased ER ROM and increased GIRD in SSE group. Contradictory.

• Page 18, Table 4: "No significant effect" column empty for 7 outcomes. Why? (Only 4 studies for function). Misleading.

• Table 5 and results: Sport performance outcomes are based on only two RCTs (n = 64), yielding very low certainty of evidence (Table 5). However, the manuscript concludes that the intervention ‘is recommended for athletes,’ which contradicts GRADE guidance

5. Discussion & Conclusion (Pages 19–23)

• Page 22, Line 5: "Structured scapular-focused exercise can be considered" – irresponsible with very low certainty for performance. Reference: GRADE Handbook – recommendations prohibited with very low certainty.

• There is no subgroup or sensitivity analysis exploring heterogeneity. The authors could have examined, for example, RCT-only studies, overhead-sport athletes only, or type-I interventions only. At present, no investigation of heterogeneity has been conducted, which limits the interpretability of the findings.

• Future directions suggest "standardized protocols" – but authors provide none.

Reviewer #2: The manuscript addresses an important and clinically relevant question regarding the impact of movement-based exercise interventions on sports performance outcomes in athletes with scapular dyskinesis. The topic is timely, given the increasing interest in scapular-focused rehabilitation and the need for evidence-based guidelines for overhead athletes. The review is well-structured, follows PRISMA methods, and provides a comprehensive narrative synthesis. However, several methodological, conceptual, and reporting limitations, particularly those affecting internal validity, transparency, and interpretability, should be addressed before the manuscript can be considered for publication.

Comments

1. In terms of clarity, in the introduction the rationale for specifically linking movement-based exercise to sport-specific performance is not sufficiently developed. Much of the introduction focuses on clinical outcomes (pain, ROM, disability), whereas the objective emphasizes sports performance metrics. The conceptual pathway from dyskinesis → biomechanical impairment → decreased performance should be more explicitly synthesized from current literature. Add a paragraph that explicitly connects scapular mechanics to performance metrics such as throwing velocity, power, stability, and proprioception.

2. The search strategy is extensive, but the PICOS criteria reveal conceptual issues. Outcome criteria requiring both pain and performance measures may have excluded relevant RCTs. Inclusion of acute and long-term interventions introduces major heterogeneity. Clarify whether studies needed both pain and performance outcomes and justify inclusion of single‑session studies.

3. Although RoB‑2 and ROBINS‑I were applied, several inconsistencies are present. Some studies lacking allocation concealment or blinding are rated as low measurement bias despite subjective outcomes. Provide more detailed justification for each bias domain and acknowledge the high likelihood of performance/detection bias in exercise trials.

4. The GRADE table contains inconsistencies. Moderate certainty ratings for ROM and function are questionable given small samples and heterogeneity. Reassess GRADE downgrades for risk of bias and inconsistency.

5. The decision not to provide a meta-analysis should be justified better. Several outcomes (e.g., IR ROM, ER strength, VAS) are conceptually identical and potentially meta‑analyzable. Either provide clear methodological justification or conduct small meta‑analyses.

6. Only two studies include true sport‑specific performance metrics. The Discussion extrapolates performance improvements from clinical improvements, which is speculative. I think that these should be excluded from your study since they are completely different populations and aims.

7. Sections 4.1–4.3 contain repeated explanations and mechanisms. Condense the Discussion to focus on key certainties, implications and research needs.

8. Some in‑text citations require correction to meet PLOS ONE requirements. Check again the references.

9. Address minor stylistic and grammatical issues to improve clarity.

10. The PROSPERO registration number (CRD42025635493) appears in the manuscript; however, upon inspection, the registered title “How long should athletes stretch? acute stretching durations for flexibility and performance: a systematic review and meta-analysis” is entirely unrelated to the present manuscript, which focuses on scapular dyskinesis and movement-based rehabilitation. This discrepancy raises concerns regarding protocol transparency, adherence to pre-specified methodology, and potential selective reporting. PROSPERO requires that systematic reviews follow the protocol outlined at registration or clearly justify any deviations. In this case, the topic, population, outcomes, and intervention categories differ completely from the registered protocol, suggesting that either the wrong registration number was inserted or the review did not follow a prospectively registered plan.

Recommendation: Need for Major Revision and Potential Re-structuring of Study Aims.

**Do you want your identity to be public for this peer review?** For information about this choice, including consent withdrawal, please see our For information about this choice, including consent withdrawal, please see our Privacy Policy .

Reviewer #1: No

Reviewer #2: No

---

## [Author Response · Author response to Decision Letter 1]

4 Feb 2026

Thank you for your efforts. We have now updated the manuscript and have also uploaded our response to the comments from the reviewers and editor.

---

## [Decision Letter · Decision Letter 1]

23 Feb 2026

Effects of scapular-focused movement-based exercises on sports performance of athletes with scapular dyskinesis: A systematic review

PONE-D-25-57628R1

Dear Dr. Mohammad Alghosi

We’re pleased to inform you that your manuscript has been judged scientifically suitable for publication and will be formally accepted for publication once it meets all outstanding technical requirements.

Kind regards,

Holakoo Mohsenifar

Academic Editor

PLOS One

Additional Editor Comments (optional):

Reviewers' comments:

Reviewer's Responses to Questions

**Comments to the Author**

Reviewer #1: (No Response)

2. Is the manuscript technically sound, and do the data support the conclusions?

Reviewer #1: Yes

3. Has the statistical analysis been performed appropriately and rigorously?

Reviewer #1: Yes

4. Have the authors made all data underlying the findings in their manuscript fully available?

Reviewer #1: Yes

5. Is the manuscript presented in an intelligible fashion and written in standard English?

Reviewer #1: Yes

Reviewer #1: Thank you for your thorough revisions and careful attention to the reviewers’ suggestions. The manuscript has improved substantially in clarity, organization, and presentation.

**Do you want your identity to be public for this peer review?** For information about this choice, including consent withdrawal, please see our For information about this choice, including consent withdrawal, please see our Privacy Policy .

Reviewer #1: No

---

## [Editor Report · Acceptance letter]

PONE-D-25-57628R1

PLOS One

Dear Dr. Alghosi,

I'm pleased to inform you that your manuscript has been deemed suitable for publication in PLOS One. Congratulations! Your manuscript is now being handed over to our production team.

Kind regards,

on behalf of

Dr. Holakoo Mohsenifar

Academic Editor

PLOS One